# Study on TLS Point Cloud Registration Algorithm for Large-Scale Outdoor Weak Geometric Features

**DOI:** 10.3390/s22145072

**Published:** 2022-07-06

**Authors:** Chen Li, Yonghua Xia, Minglong Yang, Xuequn Wu

**Affiliations:** 1Faculty of Land Resources Engineering, Kunming University of Science and Technology, Kunming 650093, China; lichens@stu.kust.edu.cn (C.L.); 11301070@kust.edu.cn (X.W.); 2Department of Earth Science and Technology, City College, Kunming University of Science and Technology, Kunming 650233, China; 20130051@kust.edu.cn

**Keywords:** point cloud registration, weak geometric features, multi-view convolutional neural networks, cosine similarity

## Abstract

With the development of societies, the exploitation of mountains and forests is increasing to meet the needs of tourism, mineral resources, and environmental protection. The point cloud registration, 3D modeling, and deformation monitoring that are involved in surveying large scenes in the field have become a research focus for many scholars. At present, there are two major problems with outdoor terrestrial laser scanning (TLS) point cloud registration. First, compared with strong geometric conditions with obvious angle changes or symmetric structures, such as houses and roads, which are commonly found in cities and villages, outdoor TLS point cloud registration mostly collects data on weak geometric conditions with rough surfaces and irregular shapes, such as mountains, rocks, and forests. This makes the algorithm that set the geometric features as the main registration parameter invalid with uncontrollable alignment errors. Second, outdoor TLS point cloud registration is often characterized by its large scanning range of a single station and enormous point cloud data, which reduce the efficiency of point cloud registration. To address the above problems, we used the NARF + SIFT algorithm in this paper to extract key points with stronger expression, expanded the use of multi-view convolutional neural networks (MVCNN) in point cloud registration, and adopted GPU to accelerate the matrix calculation. The experimental results have demonstrated that this method has greatly improved registration efficiency while ensuring registration accuracy in the registration of point cloud data with weak geometric features.

## 1. Introduction

The earth has a land area of 149 million square kilometers, of which 2439.63 square kilometers (or only 16.4%) are for residential and agricultural activities. Although most of the remaining land is not suitable for human survival, such as mountains, deserts, and other areas, it contains a lot of oil, ore, wood, water, and other resources. The ongoing exploitation of these areas could not be realized without the support of geological surveys and mapping. In the field, traditional survey methods and equipment, such as total station and level [1], were often time-consuming, laborious, and even dangerous. Three-dimensional laser scanning technology was born in the 1990s. It mainly consists of a laser rangefinder and a reflection prism. Characterized by non-contact and automation, it can quickly collect 3D point cloud data with high precision and density. Terrestrial Laser Scanning (TLS) uses a fixed sensor that is installed on a tripod to scan. It has been widely used in field surveys due to its advantages of being portable and having an independent power supply. Therefore, the processing of point cloud data has become a hot topic for scholars.

Point cloud registration can be used for clustering, identification, segmentation, and modeling, and it is an important part of point cloud processing. Besl [2] proposed the iterative closet point (ICP) algorithm, which calculates the optimal coordinate transformation using a least-square iteration, took the Euclidean distance between point clouds as the objective function, and achieved point cloud registration by reducing the distance between point clouds. This algorithm has been widely adopted for its high calculation accuracy and intuitive simplicity. However, it requires a high initial position of the point cloud and the global search strategy makes the registration efficiency low and easy to fall into the local optimal solution. In view of these problems, some scholars have added a coarse registration algorithm before the ICP registration, which has effectively reduced the calculation of the corresponding point search and prevented it from falling into the local optimal solution. For example, the SAC-IA [3] algorithm used the Huber loss function to quickly extract a set of transformation matrices with small errors. However, it relied on the point feature histogram, which made the calculation of the FPFH slow and reduced its efficiency when handling a large number of points. Although the point cloud is down-sampled, it causes the loss of some feature points and reduces registration accuracy.

TLS point cloud registration [4,5] could be affected by density changes, unstable overlaps, a large number of points, occlusions, noise, and other problems, which increase the difficulty of registration. In particular, in regions with weak geometric features where there are no obvious geomorphic changes, objects with obvious geometric features, such as houses and roads, cannot be set as a reference for registration, which makes TLS point cloud data registration extremely difficult.

With the development of neural networks, computers have become more powerful for performing complex calculations. There have also emerged a number of deep-learning-based methods for point cloud registration, such as in the field of 2D imaging, convolutional neural networks (CNN) are a very mature approach to image processing with the invariance of translation, rotation, and scaling. PointNet [6] and PointNet++ [7] used CNN in point cloud processing and are very effective in point cloud classification and semantic segmentation. Aoki et al. [8] have applied the classic visual Lucas–Kanade (LK) algorithm for image alignment to PointNet and improved it. The PointNetLK algorithm Aoki proposed had high accuracy in point cloud registration and performed well in the Gaussian noise test.

Multi-view convolutional neural networks (MVCNN) [9] can obtain multiple 2D rendered images from different viewpoints of 3D objects, extract 3D shape descriptors, and train them with 2D image convolutional networks. The trained model can be used in 3D object recognition [10], attitude estimation [11], and retrieval classification, and has achieved good results [12]. However, it has not been used in point cloud registration. There are several reasons for this. First, because there are a large number of noise points in the huge point cloud data, the registration work not only has a large number of calculations but also has unstable errors. The best method is to extract a small number of stable points to participate in registration, which are called key points. The NARF algorithm [13] can extract the edge contour features of objects because compared with other points, edge points are easier to be observed repeatedly and are stable. The SIFT algorithm [14] is invariant to rotation, scaling, and brightness changes, and stable to view angle changes, affine transformations, and noise. It is very suitable for key point extractions in multi-view situations. Second, the fundamental purpose of MVCNN is to detect and recognize the target and it has a limited perspective map for collecting images, which made high-precision angle adjustment impossible. Therefore, in this study, we designed a new iterative method, which can continuously collect images with finer angle changes for calculating the point cloud transformation matrix. Third, for symmetrical objects, such as cubes and spheres, the same view may be obtained from different observing angles, resulting in registration failure. However, in-the-wild scenarios, such as large-scale mountain and karst caves, the terrain is irregular with a complex and changing surface and it is impossible to obtain the same view from different angles. Therefore, we expanded the application of MVCNN into the registration of large-scale weak geometric features in TLS point clouds, though the problems of large amounts of point cloud data, high noise, and low overlap still need to be solved. The main contributions of this study are summarized as follows:We used the NARF algorithm and SIFT algorithm to extract stable and repeatable points as the important elements of point cloud registration, which overcame the problem of enormous point cloud data in field scenes.The maximum entropy theory was used to search for the source point cloud viewpoints that contained the largest amount of information to ensure that the registration work could be carried out under reasonable initial conditions.We proposed a novel iterative structure for the target plate. Twenty-five viewpoints were tested each time, and GPU-accelerated computation was invoked to rapidly narrow down the search area for the best viewpoint.We designed the point cloud multi-view convolution neural network (PC-MVCNN) model. In this model, the feature of the image matrix was extracted and the matching value was calculated using cosine similarity. The point with the highest score was recorded as the best viewpoint of this iteration and involved in the next iteration. A dynamic threshold of error was used as a limit to prevent infinite iterations.

The rest of this paper is organized as follows. Section 2 reviews recent research progress in point cloud registration. Section 3 introduces the PC-MVCNN model structure in detail. Section 4 uses multiple groups of experimental data for point cloud registration and makes a comparative analysis with the results of other algorithms. Section 5 represents the conclusions of this paper.

## 2. Related Work

Point cloud registration mainly involves point cloud filtering, feature descriptors, and a 3D surface matching algorithm. Moreover, some new methods for TLS point cloud registration will also be introduced.

### 2.1. Point Cloud Filtering

There are five traditional methods for filtering point cloud data: statistics-based, projection-based, neighborhood-based, signal-processing-based, and partial-differentiation-based. For instance, based on the surface variation factor, Jia Chaochuan [15] has divided the point cloud into flat regions and mutant regions and has used different filtering algorithms for different feature regions. Jia has used median filtering for flat regions and bilateral filtering for abrupt regions. In this way, the point cloud can be smoothed without losing the detailed features of the point cloud but it increases the computational complexity. Tuba Kurban [16] has established a filtering method based on plane fitting through the differential evolution algorithm. It was compared with the commonly used singular value decomposition method to obtain the plane parameters and to filter noisy point cloud data. However, it is not applicable for areas with unclear geometric features. Blind use of it will increase the difficulty of the registration calculation. Faisal Zaman [17] has used particle-based optimization technology to automatically approach the optimal bandwidth for multi-core density estimation, which ensured the robust performance of density estimation. Clustering technology based on mean shift can remove outliers and apply bilateral grid filtering to smooth the remaining points. It has strict requirements for manual selections of the threshold, which may lead to the excessive displacement of the point coordinates.

### 2.2. Feature Descriptor

From the perspective of scale, feature descriptors are generally divided into local features and global features. For example, the descriptions of local normal and global topological features are covered in the 3D point cloud feature description and extraction. Intrinsic shape signatures (ISS) proposed by Zhong Yu [18] have established the local coordinate system of key points and found the covariance matrix of key points and domain points. He has also used the relationship between the eigenvalues of the covariance matrix to describe the degree of the point’s feature. Rusu [19,20] has proposed the point feature histogram (PFH) descriptor and a fast point feature histogram (FPFH) descriptor. PFH could describe the k-neighborhood geometric properties of points by parameterizing the spatial differences between the query point and its neighbor points and forming a multidimensional histogram. All interactions between the estimated normal directions are considered to capture the best surface variation of a sample to describe its geometric characteristics. The theoretical computational complexity is O (nk^2^), but a large amount of computation is not suitable for the real-time system, thus FPFH further simplified FPH. FPFH reduces the computational complexity to O (nk) with good discrimination ability retained, but it does not count the calculation parameters of all the adjacent points of the fully interconnected points. Therefore, some important point pairs that could contribute to capturing geometric features around the query points may have been missed. The signature of histograms of orientations shot (SHOT) descriptor, proposed by Salti Samuele [21], has taken the advantage of the low robustness of the SIFT feature of 2D images in the neighborhood of the whole normal vector and divided the entire neighborhood into 32 volumes. However, encoding in the form of a local histogram is inevitably affected by edge effects, which need to be weakened and eliminated by taking complex interpolation measures. Both the SURF [22] algorithm and the SIFT algorithm perform linear diffusion in the Gaussian scale space but there are only great differences. SIFT uses the DOG image, whereas SURF uses the Hessian matrix determinant approximation image. It simply blurs the image and does not pay attention to the boundary of the original image, which will lead to the same degree of smoothing of the object boundary and noise. The KAZE [23] descriptor is a 2D feature detection and description method in nonlinear scale space, which can smooth the noise while preserving the original boundary of the target as much as possible. It also has good robustness in the case of information loss caused by image blur, noise interference, and compression reconstruction. BRIEF [24] describes the detected feature points, abandons the traditional method of describing feature points using a regional gray histogram, greatly accelerates the speed of establishing feature descriptors, and greatly reduces the time for feature matching. However, there are also some defects, such as scale invariance, rotation invariance, and sensitivity to noise. The ORB (Oriented FAST and Rotated BRIEF) [25] algorithm is a combination of the FAST feature point detection and the BRIEF feature descriptor. It is improved and optimized, giving the OBR features a variety of local invariance and providing the possibility for real-time computing. The BRISK [26] feature descriptor constructs an image pyramid to detect feature points in multi-scale spaces. It can achieve better matching results for blurred images. DAISY [27] is a local image feature descriptor for dense feature extraction. Similar to SIFT, DAISY uses a block statistical gradient direction histogram. The difference is that DAISY improves the blocking strategy by using Gaussian convolution to cluster the gradient direction histogram. In this way, the feature descriptor can be extracted quickly and densely using the fast computability of Gaussian convolution.

With the rapid development of neural networks, their fast and efficient computing system has also been applied in point cloud registration. Consequently, research on TLS point cloud registration in large scenarios has gradually increased. Some scholars have proposed self-learning feature descriptors. For instance, Yew [28] et al. have proposed a weakly supervised 3D Feat-Net model, which demonstrated significantly better registration performance on field point cloud datasets than the traditional descriptors. However, it still had the problem of high numbers of errors and limited the size of the input point cloud under the condition of a large amount of noise. Currently, it cannot meet the requirements of point cloud registration in complex scenarios. Andy Zeng et al. [29] have proposed a 3D match local geometric descriptor based on 3D ConvNet, which could perform well in multi-scale and low-resolution scenarios with a large amount of training, and the learning efficiency and registration accuracy of the descriptors learned through self-supervision were more prone to be affected in areas with weak geometric features.

However, it mainly extracts the features of areas with large changes, so the registration accuracy is lower in scenarios with weak geometric features.

### 2.3. TLS Point Cloud Registration

In recent years, TLS point cloud registration algorithms have been proposed. JoKDNet [30] has improved registration accuracy and robustness in a variety of field environments by improving the detection and description of key points. However, its down-sampling increases the registration difficulty of the low-overlapping point cloud, and the transformation parameters cannot be directly optimized without considering feature matching. Based on the Coherent Point Drift (CPD) algorithm, Y. Zang [31] has improved the robustness of the algorithm in environments with complex point cloud density changes by combing the covariance descriptor. However, this has not taken the influence of point density of two-point probability into account, and this could further slow down convergence efficiency. The 4PCS algorithm [32] uses the RANSAC algorithm framework to reduce spatial matching operations by constructing and matching congruent four-point pairs, thus accelerating the registration process. It is suitable for point cloud registration in scenes with small overlapping areas or large changes in overlapping areas without pre-filtering and denoising the input data. HRegNet [33] performs registration on hierarchically extracted key points and descriptors instead of using all the points in the point cloud. The overall framework combines the reliable features in the deeper layers and the precise position information in the shallower layers to achieve robust and precise registration. SpinNet [34] uses a neural feature extractor that leverages powerful point-based and 3D cylindrical convolutional neural layers are utilized to derive a compact and representative descriptor for matching.

## 3. Method

We propose a method for 3D laser point cloud data registration using a multi-view convolutional neural network (MVCNN) for the image dimensions, which consists of three steps. First, we extract the robust and repeatable key points from the point cloud in order to reduce the number of computations and improve registration efficiency. In this study, the NARF + SIFT algorithm is used to extract key points with stronger expression capabilities. Second, a PC-MVCNN model suitable for point cloud registration is proposed, and a new iterative form of convolution is designed to find the best matching viewpoint and calculate the transformation matrix. Third, the GPU is used to accelerate the operation of the matrix in the convolution.

### 3.1. Key Points Extraction

Because of the uneven surface of the study targets, even for the laser point cloud data collected on the same surface, the occlusion will change excessively with different viewpoints, leading to the phenomenon of the low repeatability of most point clouds. In addition, the calculation of all the points increases the computation workload and makes it easy for them to be affected by noise. Therefore, whether a small number of stable points can replace the original point cloud to participate in subsequent processing has become our reason for extracting the key points. The key point is an element from the original point cloud and any point in the point cloud may be used as a key point.

#### 3.1.1. Voxel Filtering

The point cloud data collected in a large scene has the following characteristics. First, their density is irregular. The point clouds close to the sensor are dense and those distant from the sensor are sparse. Second, the occlusion is serious. In the field, trees, pedestrians, vehicles, and other factors can often cause point cloud occlusion, resulting in a large number of outliers. Third, the amount of data is large. The TLS data of a single station can reach several million points in a large scenario, which requires a huge amount of computation. Voxel filtering, insensitive to density, can remove noise points and outliers to a certain extent while down-sampling the point cloud. We set the resolution for the input point cloud and divided the large cube into different smaller cubes. We used the coordinates of the centroid of the points and calculated in each small cube to approximate several points in the cube. Although we created one more step in the centroid calculation rather than using the substitution of the center, which slows down progress, this can better represent the real features of the point cloud. Voxel filtering is used to preprocess the point cloud, which can reduce the influence of noise points on registration accuracy and improves efficiency. 

#### 3.1.2. NARF Key Points

In both 2D and 3D fields, the points on the edges of objects have a strong descriptive ability, which makes them more likely to be selected as key points than other points. Unlike the edge of a 2D image with obvious grayscale changes, the edge of a point cloud has a more definite physical meaning, which represents the boundary between foreground and background objects. The NARF algorithm [13] can extract the edge key points from depth images generated from arbitrary 3D point cloud data, improve edge sensitivity, better retain surface details, and maintain its stability and repeatability in multiple perspectives. We score every point to determine how likely the point is to be an edge point. First, the edge may be in four directions from a point: up, down, left, and right. So, we calculate the relative distance between *p_i_* and the point to the right of *p_i_*. By comparing this relative distance with the given threshold, we could determine whether the edge was to the right of the point. If the distance was much larger than the given threshold, the neighboring point to the right of *p_i_* is not in the same plane as *p_i_*. As shown in Formulas (1) and (2) [13]:(1)$$p_{\mathrm{right}}=\frac{1}{m_{p}}\overset{m_{p}}{\underset{i=1}{\sum }}p_{x}+i,y$$

In order to increase the adaptability to noise, the point on the right is taken as the average of several points on the right.
(2)$$S_{\mathrm{right}}=\max\left(0,1-\frac{\delta }{d_{\mathrm{right}}}\right)$$

Based on the calculation of Formulas (1) and (2), the edge of the point cloud is obtained. For a surface formed by the point cloud, the curvature of a certain point is undoubtedly a very important structural description factor. The larger the curvature of a point, the more drastic the change in the surface at the point and the higher the probability of it becoming a feature point. On the point-cloud-transformed depth image, Principal Component Analysis (PCA) is performed on the *p_i_* point and its neighboring points. The main direction *ν* of a 3D vector could be obtained as well as the curvature value *λ*. Regarding the edge point, its weight *w* is set to 1 and *ν* is the edge direction. For other points, the weight *w* is taken as 1 − (1 − *λ*)^3^ and the direction as the projection of *ν* on the plane *p*. The plane *p* is perpendicular to the line from *p_i_* to the origin. At this point, each point has two quantities, a weight, and a direction. Substituting the weight and direction in the following Formulas (3)–(6) [13] is the probability of a point becoming a feature point.
(3)$$I_{1}\left(p\right)=\min\left(1-w_{n_{i}}\max\left(1-\frac{10\cdot \|p-n_{i}\|}{\sigma },0\right)\right)$$
(4)$$f\left(n\right)=\sqrt{w_{n}\left(1-\left|\frac{2\cdot \|p-n\|}{\sigma }-\frac{1}{2}\right|\right)}$$
(5)$$I_{2}\left(p\right)=\underset{i,j}{\max}\left(f\left(n_{i}\right)f\left(n_{j}\right)\left(1-\left|\cos\left(\alpha_{n_{i}}^{\prime }-\alpha_{n_{j}}^{\prime }\right)\right|\right)\right)$$
(6)$$I\left(p\right)=I_{1}\left(p\right)\cdot I_{2}\left(p\right)$$

Finally, maximum suppression is carried out and all points with interest values above the threshold can be regarded as NARF key points.

#### 3.1.3. SIFT Key Points

Extracting the NARF key points made it easier to extract the SIFT key points. All points in the neighborhood *r* of each NARF key point are marked and they no longer participate in the extraction of the SIFT key points. A scale-invariant feature transform (SIFT) algorithm [14], which can keep good invariance to luminance changes, noise, rotations, and shifts, can extract stable key points in the central region. The specific steps are shown in Formulas (7)–(11) [14]:

1.Generate the scale space. The scale space of an image was defined as


(7)
L(x,y,σ)=G(x,y,σ)×I(x,y)



(8)
$$G\left(x,y,\sigma \right)=\frac{1}{2\pi \sigma^{2}}e^{-\frac{\left(x^{2}+y^{2}\right)}{2\sigma^{2}}}$$


In the formula, *G*(*x*, *y*, *σ*) is a scale-variable Gaussian function: *σ* represents the scale-space factor, reflecting the degree of image blur, and (*x*, *y*) represents the coordinates of the pixel.

2.Detect the extreme points in the scale space and construct the difference of the Gaussian (DOG) function:


(9)
$$\begin{array}{ll} D\left(x,y,\sigma \right) & =\left[G\left(x,y,k\sigma \right)-G\left(x,y,\sigma \right)\right]\times I\left(x,y\right)\\ & =L\left(x,y,k\sigma \right)-L\left(x,y,\sigma \right) \end{array}$$


In the formula *D*(*x*, *y*, *σ*) is a Gaussian difference function, which is introduced to effectively detect the stable feature points in the scale space and *k* is a constant. When detecting extreme points, each pixel is compared with 26 points (8 adjacent points of the same scale and 9 × 2 adjacent points of its upper and lower scales). If the DOG operator has extreme values in these 26 neighborhoods, the point is considered a feature point.

3.Accurately locate the extreme point.

By fitting a three-dimensional quadratic function, the position and scale of feature points are precisely located whereas the low-contrast feature points and unstable edge response points are removed to enhance matching stability and improve anti-noise ability. We also perform curve fitting of the DOG function in the scale space and use the Taylor expansion of the DOG function:(10)$$D\left(X\right)=D+\frac{\partial D^{T}}{\partial X}X+\frac{1}{2}X^{T}\frac{\partial^{2}D^{T}}{\partial X^{2}}$$

In the formula, the vector *X* = (*x*, *y*, *σ*). We set the first derivative of this formula as 0 and obtain the offset vector of the exact position of the feature point:(11)$$\hat{X}=-{\frac{\partial^{2}D}{\partial X^{2}}}^{-1}\frac{\partial D}{\partial X}$$

An accurate estimate of the feature point is obtained by adding *x* to the coordinates of the original feature point.

The extracted NARF + SIFT key points are shown in Figure 1. The blue are the NARF key points, which are mostly located in edge regions with structural changes, and the red are SIFT key points, which are mostly located in structural stable regions.

### 3.2. Extract the Maximum Information Viewpoint from the Source Point Cloud

The purpose of point cloud registration is to find the XZY coordinate axis offset and rotation angle of the point cloud. In a 3D convolution, introducing more parameters will increase the dimension collapse. MVCNN uses 3D data of objects to obtain 2D renderings from different perspectives as the original training data. Current research on MVCNN focuses on the recognition and classification of 3D objects but it has not been applied in point cloud registration. The reasons are as follows: First, the perspective map collected by MVCNN is limited and its fundamental purpose is target detection and recognition, which does not require high-precision angle adjustments. Second, for symmetrical objects such as cubes and spheres, the same view may be obtained when observed from different angles, resulting in registration failure. However, for large-scale mountains, karst caves, and other field scenes, the terrain is irregular and complex with strange surfaces and weak geometric features and it is impossible to obtain the same view from different points of view of the collected point clouds. Therefore, the idea of MVCNN can be extended to the registration of large-scale TLS point cloud data with weak geometric features but the following problems must be solved:Images are paired. Only by ensuring good images are provided by the source point cloud can we pave the way for matching calculations in the early stages.The designed CNN must allow images of different sizes to ensure that the source point cloud and the target point cloud can calculate the matching degrees normally.

To unify the images collected from all viewpoints in the source point cloud to a single scale, we recursively calculate the minimum bounding sphere containing all key points. Let P and R be two subsets of point set S and satisfy MB, (P, R) represents the smallest bounding sphere with a point in R as its boundary and containing P and represents the smallest bounding sphere that point set R can determine. The Algorithm 1 for extracting the minimum bounding sphere [35] is as follows:
**Algorithm****1.** Minimum Bounding SphereCount **MB** (**P**, **R**), Returns the minimum bounding sphere1: **if P** = **∅** or **|R|** = **3**, then2: **else**3:  Select a point randomly 4:  **D**←**MB** (**P** − {**p**}, **R**);5:  **if D** exists and **p** ∉ **D**, then6:  **end if**;7: **end if**;

The depth image of point cloud data mapped from any viewpoint is unpredictable. We expect that the obtained depth image has a larger point cloud area and fewer point cloud occlusions. The viewpoint reflecting the maximum amount of point cloud data is called the best viewpoint in this study. The calculation of maximum information was first proposed by Claude Elwood Shannon in 1948 using information entropy to describe the uncertainty of information. The formula is:(12)H=−∑P(x)logp(x)
(13)$$H\left(X\right)=-\overset{n}{\underset{i=1}{\sum }}p\left(x_{i}\right)\log\left(p\left(x_{i}\right)\right)$$

The yaw angle *θ* and the pitch angle *φ* are the parameters that affect the number of events. The more complex a perspective reflects, the more information and the greater the entropy. Therefore, the viewpoint containing the most information in the source point cloud can be found by comparing the entropy values.

However, the search for maximum entropy cannot be unlimited, because the sum of all viewpoints observed on an object is a sphere, which means that the number of viewpoints is infinite and all points cannot be observed directly on the sphere. In order to observe the source point cloud uniformly, we adopt the line connecting the 62 observation viewpoints and particle points as the observation direction, in which every 30° is set as a plane between 30° and 150°, and a viewpoint is set every 30° on the plane. There are 12 viewpoints in each plane, and there are 5 planes, plus the two directions of up and down. Thus, there are 62 viewpoints in total. A simple schematic diagram of viewpoint distribution is shown in Figure 2. The left figure indicates that there are two viewpoints directly above and below the viewpoints and the rest are on the minimum bounding sphere, with a total of 5 layers. Each layer has a total of 12 viewpoints in each direction. The advantage of this distribution structure is that the angle between each observation direction and the neighboring directions is 30°, which allows us to observe the target uniformly when the number of viewpoints is limited.

According to the maximum entropy theory, the images collected from the 62 viewpoints are calculated. The entropy values obtained from each viewpoint are shown in Figure 3a. The best viewpoint (b) is the blue box *θ* = 120°, *φ* = 60°, with a value of 3.72, and the worst viewpoint (c) is the red box *θ* = 210°, *φ* = 30°, with a value of 0.25. Therefore, we put (b) into the optimal viewpoint matching calculation as the input image of the source point cloud. 

### 3.3. Optimal Viewpoint Matching

The convolution operation can capture a wider range of local topology information and improves the identification of features. However, when extracting the features of the local block, it will still limit the receptive field of features. To address it, we put the whole image into the convolution operation to cope with the resulting increase in computation. In the early stages, multiple convolution layers are used to quickly calculate the initial angle region and reduce the computation range. However, multiple convolutions on a fixed area would result in the loss of information on weak geometric characteristics. Therefore, in the later period, as the increase in matching scores slows down, we reduce the number of convolutional layers to retain more cloud details and increase the differentiation. We conduct a hierarchical adjustment of the convolution operation through an error threshold. When the threshold values after multiple iterations are all at the same level, reducing the convolution layers could make the features of iterative calculation more obvious. The PC-MVCNN model is shown in Figure 4. Details of the model structure are presented below.

#### 3.3.1. Image Matrix

Aiming to quickly narrow down the search area for the best view of the target point cloud, a preliminary prediction is required. The minimum bounding sphere is also computed for the target point cloud. The best viewpoint of the target point cloud is somewhere on the sphere. We continue to use the uniform observations of the 62 viewpoints on the target point cloud to obtain a set of viewpoint maps for the initial prediction. All the viewpoint maps are input into the convolutional neural network model as input B in Figure 4. Unlike the convolution mode of 2D images, RGB information is taken as the input parameter of the image matrix and the image matrix of fixed size is limited by the pixels. The only element in the image collected by the viewpoint is the key point. If it was input into the convolutional network only as a 2D image, the weak features of the points would be replaced by large blank areas. We use a new calculation method to obtain the image matrix, dividing the viewpoint map into a 2D grid (148 × 148) and recording the number of points in each grid, which are further used as the input elements of the image matrix to obtain a larger image matrix.

#### 3.3.2. Convolution Process

Because we use the number of key points in the grid as the parameter of the input matrix, when there are no points in the grid, the value is 0 and negative numbers do not appear in the matrix, which is equivalent to adding an activation layer of a ReLU function. The convolutional neural network adopts 4 convolution layers, 3 max-pooling layers, and 1 full connection layer. Due to the particularity of the regression problem, a larger reflection field is needed to extract the overall features. After several groups of experiments, it is found that in the early calculation of the large image matrix, the convolution kernel with a size of 5 × 5 is better than 3 × 3. Therefore, a convolution kernel with a size of 5 × 5 and a step size of 1 is used in the first and second layers of the convolution layer to quickly extract the overall features. In the third and fourth layers, a convolution kernel with a size of 3 × 3 and a step size of 1 is used to refine the extraction of weak geometric features. The max-pooling layer adopts a window with a size of 2 × 2 and a step size of 2. Finally, the feature vectors of the two groups of data to be matched are output from the full connection layer, which are Vector A and Vector B, respectively.

#### 3.3.3. Calculate the Match

The cosine similarity is used to calculate the matching degree of the target point view image and the optimal point view diagram of the source point cloud. Assuming that *A* and *B* are two N-dimensional vectors, *A* is [*A*_1_, *A*_2_, …, *A_n_*] and *B* is [*B*_1_, *B*_2_, …, *B_n_*], then the cosine similarity of *A* and *B* is
(14)$$\mathrm{Similarity}\left(A,B\right)=\frac{A\cdot B}{\left\|A\|\times \|B\right\|}=\frac{\sum_{i=1}^{n}\left(A_{i}\times B_{i}\right)}{\sqrt{\sum_{i=1}^{n}A_{i}^{2}}\times \sqrt{\sum_{i=1}^{n}B_{i}^{2}}}$$

The value is in the range of [−1,1], where −1 indicates completely different and 1 indicates completely similar. However, the TLS point cloud is usually measured station by station. The overlap level of the point cloud between two adjacent stations will affect the image matching degree. If the overlapping degree of a point cloud is 80%, the matching degree will not exceed 0.8. 

#### 3.3.4. Iterative Computations

After the above calculation, the viewpoint with the highest matching degree is found, then the optimal viewpoint of the target point cloud is within the neighborhood that contains the point and has an angle less than or equal to 15°. This area is a spherical crown according to the formula, and the surface area *S* is
(15)$$S=\int_{\theta }^{\frac{\pi }{2}}dS=\int_{\theta }^{\frac{\pi }{2}}2\pi r\cdot Rd\theta =2\pi R^{2}\int_{\theta }^{\frac{\pi }{2}}\cos\theta d\theta \;=2\pi R^{2}\left(1-\sin\theta =2\pi RH\right)$$

The iterative search on the spherical crown is different from the preliminary prediction of the minimum bounding sphere. For the convenience of calculation and illustration, we projected it onto a circular surface, where the points correspond to the points on the spherical crown one by one. Assuming that the best viewpoint P of the target point cloud is located anywhere on the circle, in order to quickly obtain it, the circle surface is regionally partitioned, as shown in Figure 5, and is shaped like a dartboard with each included angle of 30°. If it is divided into inner and outer circles at 1/2 of the radius, there are two cases for the location of point P: (1) point P is located in the outer circle and (2) point P is located in the inner circle. Each node on the target plate is used as the observation point and the spherical center is observed. The viewpoint map obtained is matched with the optimal viewpoint map of the source point cloud and the node with the highest matching degree is found, which is assumed to be A. Then a circle is made with point A taken as the center and (1/2) R as the new radius, which is used as the target plate for the next iteration, further narrowing the search. If the matching degree is calculated again and the score of A is still the highest, the operation will not stop; instead, the inner circle of the target plate of the last iteration is again used as the outer circle, and the inner circle of the target plate is regenerated at 1/2 of the radius as the new target plate. The iteration will be repeated until the matching degree no longer improves and it enters the optimal solution or has had the maximum number of iterations. Eventually, the highest score is the best matching point of view and the optimal transformation matrix is extracted.

### 3.4. GPU-Accelerated Matrix Computing

Each iteration on the target plate requires 25 image convolution computations, which obviously increases the computation load. With the rapid developments in computer hardware, the performance of CPUs using Single Instruction Single Data (SISD) and Multiple Instruction Stream Multiple Data (MIMD) in large-scale parallel computing is far inferior to GPU that adopts Single Instruction Multiple Data (SIMD). This is because the GPU has larger video memory and more operation units, such as integers, floating-point multiplication units, special operation units, etc., which enable it to maintain good performance in large-throughput calculations.

Thus, it is necessary to use GPU acceleration in convolution operations to improve computing efficiency. However, the GPU cannot run independently and data and computing tasks must be transferred between the GPU and CPU through the bus. In traditional single-cycle calculation, the cosine similarity formula is called 25 times in each iteration, which has serious redundancy. Therefore, we loaded 25 groups of data in the same iteration in the CPU and then imported them into the GPU for calculation. Each iteration only calls the cosine similarity formula once, which greatly improves computational efficiency. As shown in Table 1, we use CPU and GPU, which were published at the same time to compare the import time and the iteration time of the two site clouds. The experimental results show that there is little difference between the two devices in the speed of importing data and that the GPU is nearly 550% faster than the CPU in the speed of iterative computing.

## 4. Experiment and Analysis Results

In order to verify the performance of the proposed algorithm’s registration accuracy and speed, this algorithm is compared with the improved CPD [31] algorithm, the PointNetLK [8] algorithm, the 3D-NDT [36] algorithm, and the SAC + ICP [3] algorithm. The SAC-ICP algorithm is an improved version of the classical ICP algorithm, which adds the FPFH descriptor. It is a typical representative of traditional registration algorithms. Three-dimensional NDT does not use feature calculation and matching of corresponding points in the registration process, so the time is faster than other methods. It is often used to register point clouds with a single site number of more than 100,000 points. Finally, the PointNet method plays an important role in point cloud processing with neural networks because of its brilliant performance in point cloud processing in recent years. We choose the PointNetLK algorithm, which is a very successful application of the PointNet algorithm in point cloud registration. The improved CPD algorithm is designed specifically for TLS point cloud registration of complex scenes. Especially in complex environments with disordered vegetation or point-density variations, it has high accuracy and efficiency.

Because the purpose of this paper is to register TLS point clouds of large field scenes with weak geometric features, it should not select ground objects with obvious features such as buildings and roads. The WHU-TLS mountain [37,38,39] point cloud of Wuhan University and three sets of TLS point cloud data collected in the field are selected for the experiment, as shown in Figure 6. The experiment covers four types of common wild terrain, including mountains (a), cliff walls (b), karst caves (c), and forests (d). The specific number of stations and points of data set are shown in Table 2. The computer used in the experiment is configured with the 3.60 GHZ AMD 3500X CPU, 32G RAM, GTX3080Ti GPU, and the software is Visual Studio 2019 with the PCL1.12.0 library and C++ programming language. This section is divided into subheadings and should provide a concise and precise description of the experimental results, their interpretation, as well as the experimental conclusions that can be drawn.

The Figure 7 shows the local details of five algorithms in the point cloud registration of the WHU-TLS mountain dataset of Wuhan University. It can be seen in the figure that the PC-MVCNN algorithm retains the bump details of the mountain surface well and takes the shortest amount of time. Although the improved CPD algorithm has a small registration error, it has an obvious point cloud overlap leading to the problem of double images. The 3D-NDT algorithm focuses the calculation on the region with obvious features and does not control the matching error of the point cloud in remote areas without the surface details extracted. Moreover, the large number of calculations leads to an increase in registration time. Both the PointNetLK algorithm and SAC + ICP algorithm have obvious point cloud migration problems. Table 3 lists the specific registration times and errors.

As shown in Figure 8, the point cloud data of cliff walls are characterized by a wide view, few obstructions, a high degree of overlap between adjacent sites, and non-adjacent sites also have a high overlap area. The overall cliff tends to be stable but with rich surface texture details, which is a typical large-scale point cloud dataset with weak geometric features that can be used to test the matching ability of the registration algorithm in a good environment. The registration results are shown in Table 4. The PC-MVCNN algorithm is faster with the best preservation of surface information. The improved CPD algorithm has a good matching performance overall, but in terms of local details, the point cloud has no surface-texture information. The PointNetLK algorithm shows reduced computational efficiency for a large range of point cloud data, which is related to its computational complexity. The SAC + ICP algorithm slows down the calculation of the FPFH descriptor due to the increase in the number of points and the rich detailed features, which obviously slows down registration efficiency.

The karst cave point cloud data is obtained by station scanning from the entrance to the exit. The tunnel passage of the cave is long and narrow, resulting in numerous stations and points, and there is a low degree of overlap between adjacent point clouds, which can be used to measure the matching ability of the registration algorithm for low-overlapping point clouds. As shown in Figure 9, this area is a corner in the middle of the karst cave and the point cloud overlap is the lowest at the corner, which can best reflect the differences in registration accuracy. The registration results are shown in Table 5. The PC-MVCNN algorithm uses the cosine similarity to score the image-matching degree. If the overlap degree was high, the overall score would be high, and vice versa. Only the point with the highest score is taken as the best viewpoint, so it is not sensitive to the overlap degree of the point cloud and it can still maintain good registration accuracy when the overlap degree is low. The improved CPD algorithm has the problem of misregistration due to the reduced overlap degree. The 3D-NDT algorithm does not use feature calculation and matching of corresponding points, so it is faster than other methods. However, error accumulation appears before reaching the inflection point, resulting in point cloud deviations.

Most of the point cloud data of the Haikou Forest are trees, with lots of occlusions and the most noise points, so it can be used to test the robustness of the registration algorithm against noise. The local details of the forest point cloud registration are shown in Figure 10. The PC-MVCNN algorithm has gone through voxel filtering and key point extraction in the early stages, with most noise points removed. In addition, convolution is adopted to eliminate the impact of noise, so its robustness is improved and its registration fit of the tree point cloud is very good. The improved CPD algorithm is suitable for complex environments with obvious density changes but its convergence efficiency is low. Although the 3D-NDT algorithm can use three-line interpolation to smooth the discontinuity of adjacent grids and improve robustness, the interpolation leads to an increase in registration time and slows down registration efficiency. Table 6 lists the specific registration times and errors.

## 5. Conclusions

For large-scale outdoor scenes with weak geometric features, due to the large number of point clouds, serious occlusions, large amounts of noise, and other problems, how to quickly, accurately, and retain more surface details is an important goal of point cloud registration. In this paper, a combination of the NARF algorithm and SIFT algorithm is used to extract efficient and stable key points. The mature and fast image-processing technology of MVCNN is extended to 3D laser point cloud processing, and the PC-MVCNN point cloud registration model is proposed. A novel iteration of the target plate is designed to solve the problem of the limited viewpoints in MVCNN. The fast feature extraction and registration of weak geometric feature point clouds are realized by adjusting the convolution operation of the error threshold in real time. Compared with the other four registration algorithms, the experimental results show that the PC-MVCNN algorithm not only has higher accuracy than the other three algorithms in the registration of mountain, cliff, karst cave, and forest datasets but also has good robustness in complex environments with strong noise such as forests. Moreover, in terms of efficiency, it can meet the rapid registration of a large number of point clouds because of the GPU-accelerated matrix operation. The registration of the algorithm is affected by the number of iterations and its efficiency and accuracy increase with the increase in iterations. However, the threshold of some parameters still needs further optimization and these problems will continue to be improved in future studies.

## Figures and Tables

**Figure 1 sensors-22-05072-f001:**
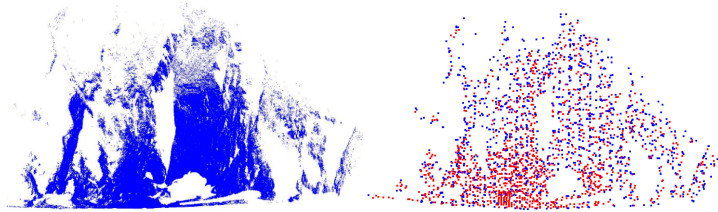
Key points extraction of NARF + SIFT algorithm.

**Figure 2 sensors-22-05072-f002:**
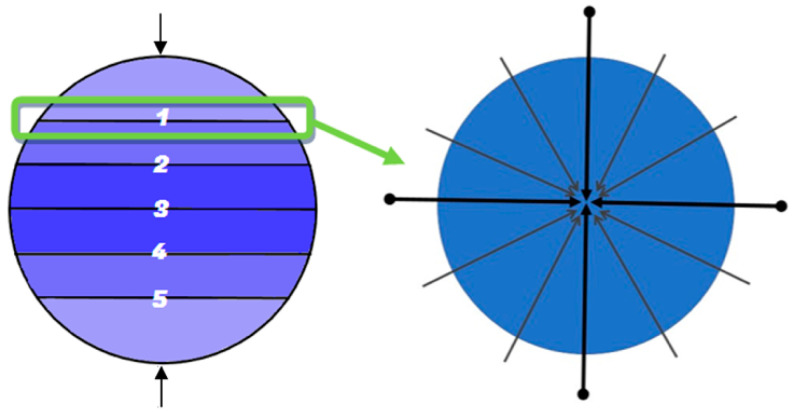
Schematic diagram of viewpoint distribution of source point cloud.

**Figure 3 sensors-22-05072-f003:**
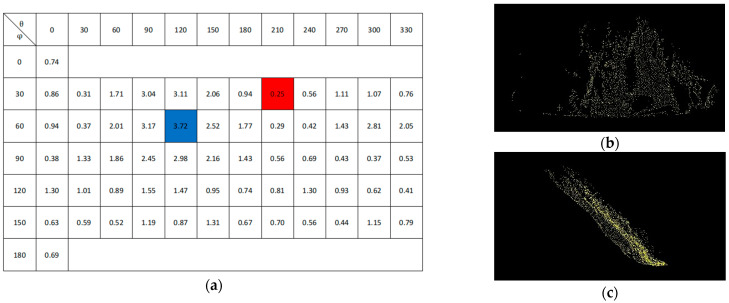
Best viewpoint selection: (**a**) entropy comparison diagram; (**b**) Best viewpoint; (**c**) Worst viewpoint.

**Figure 4 sensors-22-05072-f004:**
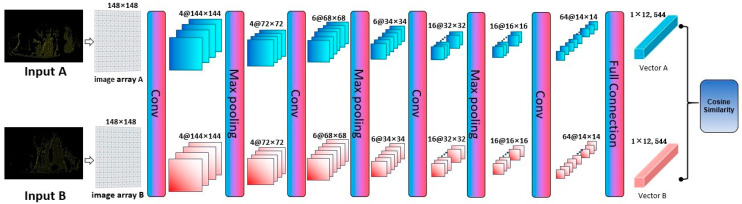
Structure diagram of convolutional neural network.

**Figure 5 sensors-22-05072-f005:**
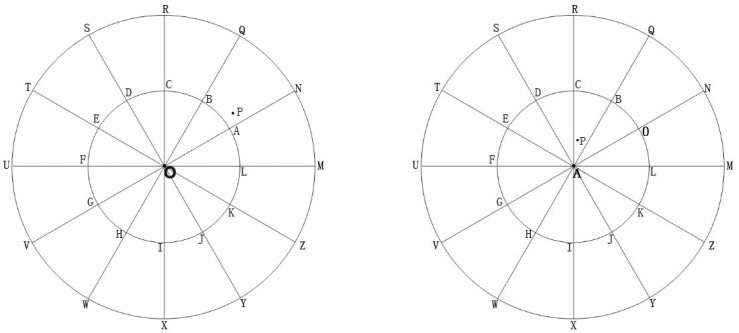
Viewpoint target plate design drawing.

**Figure 6 sensors-22-05072-f006:**
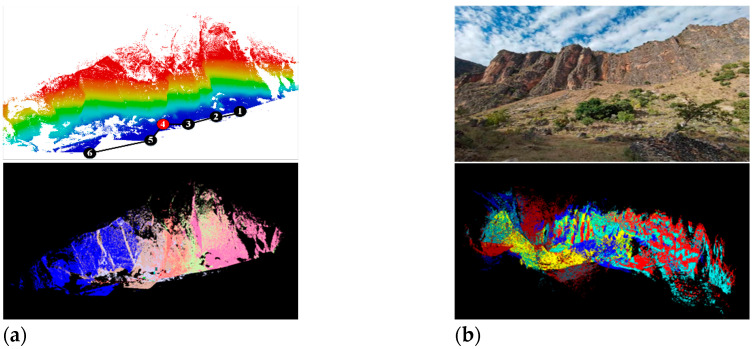
Point cloud dataset used for the experiment. (**a**) WHU-TLS Mountain dataset of Wuhan University; (**b**) Point cloud data of the Cliff Wall in Huidong County, Sichuan Province; (**c**) Point cloud data of Tangna karst cave, Bamei Town, Yunnan Province; (**d**) Point cloud data of Haikou Forest, Kunming City, Yunnan Province.

**Figure 7 sensors-22-05072-f007:**
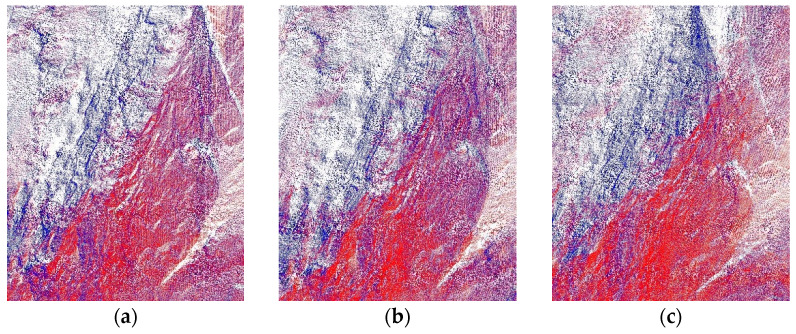
Local details of point cloud registration of WHU-TLS mountain dataset with different algorithms. (**a**) PC-MVCNN; (**b**) Improved CPD; (**c**) PointNetLK; (**d**) 3D-NDT; (**e**) SAC + ICP.

**Figure 8 sensors-22-05072-f008:**
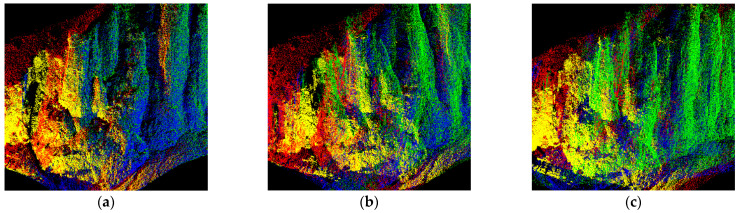
Local details of point cloud registration of the cliff wall in Huidong County, Sichuan Province. (**a**) PC-MVCNN; (**b**) Improved CPD; (**c**) PointNetLK; (**d**) 3D-NDT; (**e**) SAC + ICP.

**Figure 9 sensors-22-05072-f009:**
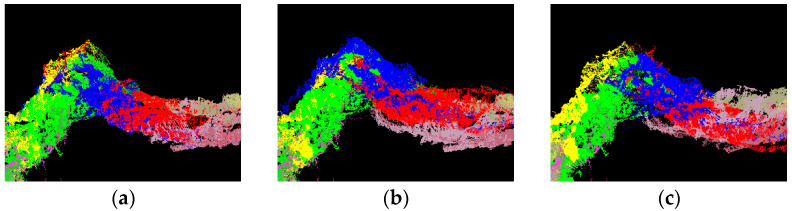
Local details of point cloud registration of Tangna karst cave, Bamei Town, Yunnan Province. (**a**) PC-MVCNN; (**b**) Improved CPD; (**c**) PointNetLK; (**d**) 3D-NDT; (**e**) SAC + ICP.

**Figure 10 sensors-22-05072-f010:**
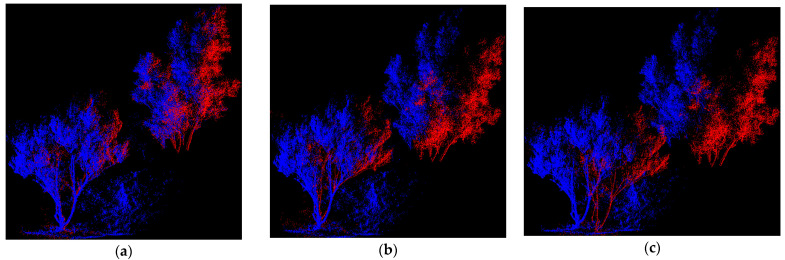
Local details of point cloud registration of Haikou Forest, Kunming City, Yunnan Province. (**a**) PC-MVCNN; (**b**) Improved CPD; (**c**) PointNetLK; (**d**) 3D-NDT; (**e**) SAC + ICP.

**Table 1 sensors-22-05072-t001:** CPU and GPU speed comparison.

Device	Import Data Time (s)	Iteration Time (s)
CPU: Platinum 8358Q	17.91	1.412
GPU: GTX 3080Ti	18.42	0.257

**Table 2 sensors-22-05072-t002:** Description of experimental data.

Point Cloud Data	Scan	Point Numbers
WHU-TLS Mountain [37,38,39]	6	19,612,517
Cliff Wall	8	14,372,836
Tangna Karst Cave	21	45,164,268
Haikou Forest	4	9,615,756

**Table 3 sensors-22-05072-t003:** Registration results of various algorithms in WHU-TLS mountain dataset of Wuhan University.

Registration Algorithm	Registration Time (s)	RMSE (m)
PC-MVCNN	78.312	0.035
Improved CPD [31]	120.454	0.058
PointNetLK [8]	114.630	0.087
3D-NDT [36]	155.718	0.064
SAC + ICP [3]	209.112	0.096

**Table 4 sensors-22-05072-t004:** Registration results of various algorithms for cliff point cloud data in Huidong County, Sichuan Province.

Registration Algorithm	Registration Time (s)	RMSE (m)
PC-MVCNN	104.506	0.041
Improved CPD [31]	177.622	0.065
PointNetLK [8]	210.815	0.094
3D-NDT [36]	204.227	0.075
SAC + ICP [3]	231.855	0.091

**Table 5 sensors-22-05072-t005:** Registration results of various algorithms for Tangna karst cave point cloud data in Bamei Town, Yunnan Province.

Registration Algorithm	Registration Time (s)	RMSE (m)
PC-MVCNN	445.912	0.080
Improved CPD [31]	514.363	0.097
PointNetLK [8]	691.705	0.284
3D-NDT [36]	556.711	0.352
SAC + ICP [3]	627.316	0.211

**Table 6 sensors-22-05072-t006:** Registration results of various algorithms for Haikou Forest, Kunming City, Yunnan Province.

Registration Algorithm	Registration Time (s)	RMSE (m)
PC-MVCNN	50.181	0.072
Improved CPD [31]	79.327	0.090
PointNetLK [8]	82.846	0.164
3D-NDT [36]	161.364	0.324
SAC + ICP [3]	102.413	0.367

## Data Availability

Not applicable.

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
