# Peer review of "Study on TLS Point Cloud Registration Algorithm for Large-Scale Outdoor Weak Geometric Features"

_sensors, 2022, doi:10.3390/s22145072_

Round 1
Reviewer 1 Report
Dear Authors,
I have reviewed the paper entitled "Study on TLS point cloud registration algorithm for large-scale outdoor weak geometric features." This paper attempts to prepare an algorithm for terrestrial laser scanning point clouds registration. Although the topic of the study could be interesting, some parts of the research are not conducted/described correctly. Please find the detailed justifications below.
1) The introduction section does not provide a description of the current state of the research field in the registration of terrestrial laser scanning point clouds. Almost a half of the introduction is about topics that are not related to point cloud registration.
a) Lack of description of methods currently used.
b) Only old measuring methods were pointed out when the work's purpose and significance were described. Authors are putting currently used methods aside to strengthen the justifications of the topic.
2) Section 2 describes related works, but in all cases, the authors state that these resolutions are "not suitable for large-scale point cloud registration," while there are no experiments and references that show these statements' background.
3) In section 2.2, the Authors described different algorithms but omitted the most used ones, i.e., SURF, ORB, KAZE, BRISK, DAISY, and their affine variants.
4) Section 2.4, "The point cloud obtained by TLS scanning is characterized by a large amount of data and much noise, which does not allow the application of traditional ICP algorithm for registration." There is no evidence for that statement, while most known software for TLS point cloud registration (i.e., Leica Cyclone) uses the ICP algorithm or its modifications. Moreover, because of its low noise level, TLS is used in research as a comparable method for MLS and other methods. The main issues related to the ICP are: "ICP is highly dependent on a good initialization step in order to prevent the algorithm becoming trapped by a local optimum. Second, incorrect closest point correspondences are very common in the registration process due to the local optimal correspondence matching strategy." [1]
5) Equations are the first usage case or were used before; if yes, add references.
6) Figure 6 and Table 1 contain apparent facts that are not related to the paper's topic.
7) There is no description of motivations for using the "Improved CPD algorithm, the PointNetLK algorithm, the 3D-NDT algorithm and the SAC+ICP algorithm". There are more algorithms used for registration (described in the literature review [1], which describes the used dataset by the authors), i.e., 4PCS, Multiview point cloud registration, and deep approaches, so the paragraph which explains why the authors chose those must be added. Moreover, there are no references for those algorithms, so the potential reader will not know what version, modification, and implementation of these algorithms were used. According to the guidelines, materials and methods "should be described with sufficient detail to allow others to replicate and build on published results."
8) There is a lack of description of the datasets. Authors should briefly describe the algorithm of dataset choosing (why these datasets were chosen, not another).
9) There is a lack of references to the used datasets!
10) What data were used as ground truth to compute the RMS values. While the dataset was not acquired in the same period, some changes may occur, i.e., Mountain dataset – 4 scans in March (not in 1 day) and 2 scans in August.
11) According to the Federal Geographic Data Committee's Geospatial Positioning Accuracy Standards Part 3, the accuracy testing by an independent source of higher accuracy (another method) is the preferred test for positional accuracy. Of course, the independent source of higher accuracy shall be the highest accuracy feasible and practicable to evaluate the accuracy of the dataset. Taking into account the type of the measurements and computations (point cloud registration – it is understood as positional accuracy expressed with RMSE), which were used in the experiment, as the reference for the performed test, a few static terrestrial laser scanning stations with an additional reference (i.e., Ground Control Network) should be used to check the RMS after registration independently. Furthermore, the authors used scan stations acquired with the time shift, so the results should be divided by with/without time shift acquisition. The lack of this part in the paper is probably the results of the introduction section, which does not include the current state of the research field.
Sincerely,
Reviewer
[1]: Dong, Z., Liang, F., Yang, B., Xu, Y., Zang, Y., Li, J., Wang, Y., Dai, W., Fan, H., Hyyppä, J., & Stilla, U. (2020). Registration of large-scale terrestrial laser scanner point clouds: A review and benchmark. ISPRS Journal of Photogrammetry and Remote Sensing, 163, 327–342. https://doi.org/10.1016/j.isprsjprs.2020.03.013
Author Response
Dear reviewer:
Thank you for your professional suggestions. We have carefully revised the paper according to your comments. Due to the length, please refer to the attachment for details.

Reviewer 2 Report
This manuscript focused on the TLS point cloud registration algorithm for large-scale outdoor weak geometric features. In general, the manuscript gives a good description of the proposed methodology, and the data analysis and results are fully discussed. The detailed comments:
1. It is noted that your manuscript needs careful editing by someone with expertise in technical English editing paying particular attention to English grammar, spelling, and sentence structure so that the goals and results of the study are clear to the reader.
2. The Figures are not clear enough, it is recommended to increase the resolution of some pictures or conduct a more effective comparison.
Author Response
Dear reviewer:
Thank you for your professional suggestions. We have carefully revised the paper according to your comments. Please refer to the attachment for details.

Reviewer 3 Report
Nice written paper, with an understandable introduction and related work presented. The results are clearly explained.
I propose to check references some abbreviations are written with lower letters.
The "Figure Figure" appears in a text when you put a reference to figures.
Author Response

(The authors gave the same response as above.)

Round 2
Reviewer 1 Report
Dear Authors,
I have re-reviewed the paper entitled "Study on TLS point cloud registration algorithm for large-scale outdoor weak geometric features." The revised text is much better than the previous one. After reading the manuscript, I only have some comments that I feel can be easily addressed.
1) The text should clearly state why the authors chose SIFT over other algorithms (described in the introduction).
2) Authors should extend the conclusion section by the main results (i.e., accuracies), as it is a self-standing part of the paper.
3) The text should be spell-checked because some omissions occurred, i.e., Line 160 "SUFT" ->SURF; double spaces, etc.
Sincerely,
Reviewer
Author Response
Thank you for your comments. We have completed the modification according to your suggestions. Please see the attachment for details.
